# Weights Having Stable Signs Are Important: Finding Primary Subnetworks and Kernels to Compress Binary Weight Networks

## Abstract

Binary Weight Networks (BWNs) have significantly lower computational and memory costs compared to their full-precision counterparts. To address the non-differentiable issue of BWNs, existing methods usually use the Straight-Through-Estimator (STE). In the optimization, they learn optimal binary weight outputs represented as a combination of scaling factors and weight signs to approximate 32-bit floating-point weight values, usually with a layer-wise quantization scheme. In this paper, we begin with an empirical study of training BWNs with STE under the settings of using common techniques and tricks. We show that in the context of using batch normalization after convolutional layers, adapting scaling factors with either hand-crafted or learnable methods brings marginal or no accuracy gain to final model, while the change of weight signs is crucial in the training of BWNs. Furthermore, we observe two astonishing training phenomena. Firstly, the training of BWNs demonstrates the process of seeking primary binary sub-networks whose weight signs are determined and fixed at the early training stage, which is akin to recent findings on the lottery ticket hypothesis for efficient learning of sparse neural networks. Secondly, we find binary kernels in the convolutional layers of final models tend to be centered on a limited number of the most frequent binary kernels, showing binary weight networks may has the potential to be further compressed, which breaks the common wisdom that representing each weight with a single bit puts the quantization to the extreme compression. To testify this hypothesis, we additionally propose a binary kernel quantization method, and we call resulting models Quantized Binary-Kernel Networks (QBNs). We hope these new experimental observations would shed new design insights to improve the training and broaden the usages of BWNs.

## 1 Introduction

Convolutional Neural Networks (CNNs) have achieved great success in many computer vision tasks such as image classification (Krizhevsky et al., 2012), object detection (Girshick et al., 2014) and semantic segmentation (Long et al., 2015). However, modern CNNs usually have large number of parameters, posing heavy costs on memory and computation. To ease their deployment in resource-constrained environments, different types of neural network compression and acceleration techniques have been proposed in recent years, such as network pruning (Han et al., 2015; Li et al., 2017), network quantization (Hubara et al., 2016; Rastegari et al., 2016; Zhou et al., 2016), knowledge distillation (Ba & Caruana, 2014; Hinton et al., 2015), efficient CNN architecture engineering and searching (Howard et al., 2017; Zhang et al., 2018b; Zoph & Le, 2017).

Comparatively, network quantization is more commercially attractive as it can not only benefit specialized hardware accelerator designs (Sze et al., 2017), but also can be readily combined with other techniques to get further improved compression and acceleration performance (Mishra & Marr, 2018; Han et al., 2016; Zhou et al., 2017). Quantization methods aim to approximate full-precision (32-bit floating-point) neural networks with low-precision (low-bit) ones. In particular, the extremely quantized models called Binarized Neural Networks (BNNs) (Courbariaux et al., 2015; 2016; Rastegari et al., 2016) which force the weights or even weights and activations to have 1-bit values ($+1$ and $-1$), bringing $32\times$ reduction in model size and making costly 32-bit floating-point

multiplications can be replaced by much cheaper binary bit-wise operations. Because of this, how to train accurate BNNs either in a post-training manner or in a training from scratch manner has attracted increasing attention. However, training BNNs poses a non-differentiable issue as converting full-precision weights into binary values leads to zero gradients. To combat this issue, most existing methods use the Straight-Through-Estimator (STE). Although there are few attempts (Achterhold et al., 2018; Chen et al., 2019; Bai et al., 2019; Hou et al., 2017) to learn BNNs without STE by using proximal gradient methods or meta-learning methods, they suffer from worse accuracy and heavier parameter tuning compared to STE based methods. In STE based methods, full-precision weights are retained during training, and the gradients *w.r.t.* them and their binarized ones are assumed to be the same. In the forward pass of the training, the full-precision weights of the currently learnt model are quantized to binary values for predication loss calculation. In the backward pass, the gradients *w.r.t.* full-precision weights instead of binary ones are used for model update. To compensating for drastic information loss and training more accurate BNNs, most state of the art STE based methods follow the formulation of (Rastegari et al., 2016) in which the binary weights are represented as a combination of scaling factors and weight signs to approximate 32-bit floating-point weight values layer-by-layer, yet also present a lot of modifications. These modifications include but are not limited to expanding binary weights to have multiple binary bases (Lin et al., 2017; Guo et al., 2017), replacing hand-crafted scaling factors with learnable ones (Zhang et al., 2018a), making an ensemble of multiple binary models (Zhu et al., 2019), searching high-performance binary network architectures (Kim et al., 2020), and designing improved regularization objectives, optimizers and activation functions (Cai et al., 2017; Liu et al., 2018; Helwegen et al., 2019; Martinez et al., 2020).

There are also a few works, trying to make a better understanding of the training of BNNs with STE. In (Alizadeh et al., 2019), the authors evaluate some of the widely used tricks, showing that adapting learning rate with a second-moment optimizer is crucial to train BNNs with STE based methods while other tricks such as weights and gradients clipping are less important. Bethge et al. (2019) shows the commonly used techniques such as hand-crafted scaling factors and custom gradients are also not crucial. Sajad et al. (2019) demonstrates learnable scaling factors combined into a modified sign function can enhance the accuracy of BNNs. Anderson & Berg (2018) makes an interpretation of why binary models can approximate their full-precision references in terms of high-dimensional geometry. Galloway et al. (2018) validates that BNNs have surprisingly improved robustness against some adversarial attacks compared to their full-precision counterparts. In this paper, we revisit the training of BNNs, particularly Binary Weight Networks (BWNs) with STE, but in a new perspective, exploring structural weight behaviors in training BWNs.

Our main contributions are summarized as follows:

- We use two popular methods (Rastegari et al., 2016) and (Zhang et al., 2018a) for an empirical study, showing both hand-crafted and learnable scaling factors are not that important, while the change of weight signs plays the key role in the training of BWNs, under the settings of using common techniques and tricks.

- More importantly, we observe two astonishing training phenomena: (1) the training of BWNs demonstrates the process of seeking primary binary sub-networks whose weight signs are determined and fixed at the early training stage, which is akin to recent findings of the lottery ticket hypothesis (Frankle & Carbin, 2019) for training sparse neural networks; (2) binary kernels in the convolutional layers (Conv layers) of final BWNs tend to be centered on a limited number of binary kernels, showing binary weight networks may has the potential to be further compressed. This breaks the common understanding that representing each weight with a single bit puts the quantization to the extreme compression.

- We propose a binary kernel quantization method to compress BWNs, bringing a new type of BWNs called Quantized Binary-Kernel Networks (QBNs).

## 2    AN EMPIRICAL STUDY ON UNDERSTANDING BWNS' TRAINING

In this section we will briefly describe BWNs we use in experiments, implementation details, scaling factors in BWNs, full-precision weight norm, weight sign, and sub-networks in BWNs.

## 2.1 DIFFERENT BINARY WEIGHT NETWORKS

BWNs generally represents those networks with binary weights, and there are several different BWNs existing. Overall they use $\alpha B$ to replace full-precision weight $W$, where $B = sign(W)$ and $\alpha$ is proposed to minimize $||\alpha B - W||$ in an either learnable or calculated way. In following experiments, we use the one implemented in XNor-Net (Rastegari et al., 2016) and denote it as XNor-BWN, and the one implemented in LQ-Net (Zhang et al., 2018a) and denote it as LQ-BWN which is 1-bit weight 32-bit activation version of LQ-Net. Other popular BWN methods like DoReFa-Net and BinaryConnect are similar to these two methods. Both XNor-BWN and LQ-BWN use STE framework, and XNor-BWN uses hand-crafted calculated scaling factors, and LQ-BWN uses learnable scaling factors.

## 2.2 IMPLEMENTATION DETAILS AND NOTATION

**Quantization:** We directly use open source codes of BWN released by authors, including XNor-BWN[1] and LQ-BWN[2].

**Dataset and Network Structure:** CIFAR-10 (Krizhevsky & Hinton, 2009) and ImageNet (Russakovsky et al.) are used in our experiments. We use VGG-7 (Simonyan & Zisserman, 2015) and ResNet-20 (He et al., 2016) on CIFAR-10, and ResNet18 on ImageNet. The strcuture is the same as original ones.

**Hyper-parameters:** We use the same training parameters on each network. The network is trained for 200 epochs. The learning rate is set initially to 0.02 and divided by 10 at 80 and 160 epochs. For random crop, we first use zero pad to resize the image into $40 \times 40$, and random crop into $32 \times 32$. For BWN trained on ImageNet, each is trained for 100 epochs. The initial learning rate is 0.1 and decays 0.1 at 30, 60, 90 epoch. The image is rescaled into $256 \times 256$ and then randomly cropped into $224 \times 224$. No additional data augmentations are used. For all networks, weight decay is applied to all Conv layers set to $4 \times 10^{-5}$.

**Notations:** In figures and tables, we will use the following abbreviations for clearer expression. BN: BatchNormalization, LR: Learning Rate. WD: Weight Decay. SF: Scaling Factors. FP: Full-precision. VGG-7 XNor-BWN: a VGG-7 network using the binarization algorithm of XNor-BWN. ResNet-20 Baseline: a full-precision ResNet-20 only using data augmentation and weight decay without any additional tricks. Other network structures with certain methods are similar to this. Large weights, large magnitude weights, and weights with larger norm have the same meaning to indicate those weights having relatively large absolute values.

## 2.3 SCALING FACTORS

According to previous methods, scaling factors are one essential element in obtaining BWNs. However, according to our experiments and analysis, we find scaling factors are not so important in training BWNs, and they can somehow be ignored without the drop in performance. Here we list four reasons to explain why scaling factors are unimportant.

**A simple proof:** BN is a common practice to be used in training BWNs. It contains two operations, **Normalization** and **Affine** as shown in Equation1. $\gamma$ and $\beta$ are the affine parameters used in BN. $\epsilon = 5e - 4$ is used in PyTorch to avoid the error of dividing zero. We use a simple proof to demonstrate that BN can absorb scaling factors as shown in Equation2. This is correct during training when one scaling factor is applied to each output channel under the **Conv-BN-ReLU** structure.

$$x' = \text{Normalize}(x) = \frac{x - \bar{x}}{\sqrt{\sigma^2 + \epsilon}} \qquad y = \text{Affine}(x') = \gamma x' + \beta \tag{1}$$

$$y_\alpha = \gamma \frac{\alpha x - \alpha \bar{x}}{\sqrt{\alpha^2 \sigma^2 + \epsilon}} + \beta \approx \gamma \frac{x - \bar{x}}{\sqrt{\sigma^2 + \epsilon}} + \beta = y \tag{2}$$

---

[1] We use the codes of DoReFa-Net to realize XNor-BWN which is the same as the original implementation. https://github.com/tensorpack/tensorpack/tree/master/examples/DoReFa-Net

[2] LQ-BWN is the 1-bit weight 32-bit activation version of LQ-Nets. https://github.com/microsoft/LQ-Nets

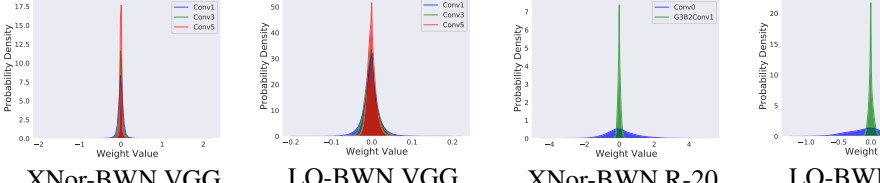

Figure 1: The visualization of full precision weights distribution in BWNs. The X-axis indicates the full precision weight value while Y-axis indicates the frequency that the value appears in a certain layer of a binary weights network. In the figure captions, VGG is VGG-7, and R-20 is ResNet-20. For VGG-7, we draw 2nd, 4th, 6th Conv layers' weight distributions (Conv1, Conv3, Conv5). For ResNet-20, we display the first and the last Conv layers' weight distributions.

**Experimental Results:** In the second aspect, we directly go to experimental results. As shown Table.2 in Appendix.B, we train different networks with and without scaling factors. The test accuracy on Cifar-10 and validation accuracy on ImageNet do not show a large difference between the two methods. Later we fix scaling factors of all layers to a certain value and magnify their learning rate according to the fixed scaling factors' magnitude. The performance does not change when fixing scaling factors. Thus, we conclude with proper learning rate, scaling factors are not essential to train BWNs.

**Compare learnable SF and $\gamma$ in BN:** LQ-BWN uses channel-wise scaling factors. From the experiments in Appendix.C, we find that these channel-wise scaling factors having a high correlation with $\gamma$ in the BN after corresponding binary Conv. This finding indicates that BN's $\gamma$ can replace channel-wise SF to some extent.

**Quantization Error Curve:** Another purpose using scaling factors is to reduce the quantization error between full-precision weights and binary weights according to a BNN survey (Qin et al., 2020). By using experiments in Appendix.D we prove that the quantization error is not actually reduced by scaling factors, but weight decay helps on this reduction.

## 2.4 Weight Norm, Weight Sign, and Sub-Networks in BWNs

We already analyse one essential element, scaling factors, in the previous section, and another essential element of BWNs are weights' signs. In deterministic binarization methods, full-precision weights' signs decide their binary weights' signs using a $sign()$ function. In this section, we will discuss the relationship between weight norm, weight sign and how to find primary binary sub-networks in BWNs.

**Weight Histogram:** We visualize the full-precision weight distribution in different layers of different networks as shown in Figure.1. Different from a bi-modal distribution, it shows a centered distribution around 0. This again proves that the actual distance or so-called quantization error is very large. And there are many weights close to zero behaving very unstable, which will change their signs with little perturbations. More experiments and visualizations are in Appendix.E.

**Flipping Weights' Signs:** We flip the weights' signs during the inference section according to weights' full-precision norm as shown in Figure.12 of Appendix.G. We flip those weights with the largest norm and the smallest norm in two experiments. It shows that even the weights have the same norm after binarization, and the changed norm is the same for the same flipping percentage, there is still a very large gap between the two results. Flipping those weights with large full-precision magnitude will cause a significant performance drop compared to those weights close to zero. This reveals that weights are different where some with small norm can tolerate sign flipping, and those with large norm cannot suffer from changing signs, even though both two kinds of weights have the same norm after binarization.

**Tracing Large Weights** From the last experiment, we conduct that weights with large norm are vulnerable and important during inference, however, the function of them during training remains unclear. Then we conduct two experiments to tracing these large weights during training. We also use "these large weights" to indicate these weights having the larger magnitude/larger norm in the

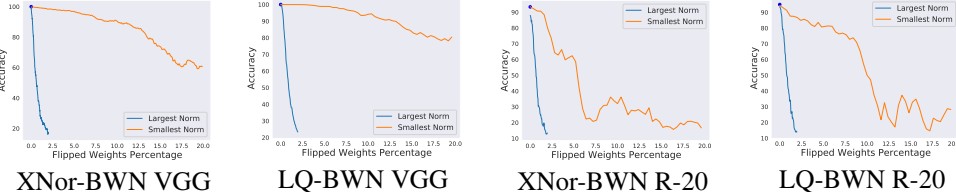

XNor-BWN VGG    LQ-BWN VGG    XNor-BWN R-20    LQ-BWN R-20

Figure 2: Inference accuracy on training sets after flipping a certain percentage of weights' signs. We design two flipping methods, flipping those weights with larger norm (from the largest norm to the smallest norm) and flipping those weights with the smaller norm. The X-axis indicates how many percentage of weights is flipped, while the Y-axis indicates the inference accuracy. The top-left point in each figure is the un-flipped case which is the same as the result reported in Table.2. This flipping operation is done to each binary Conv layer and each layer has the same flipping percentage.

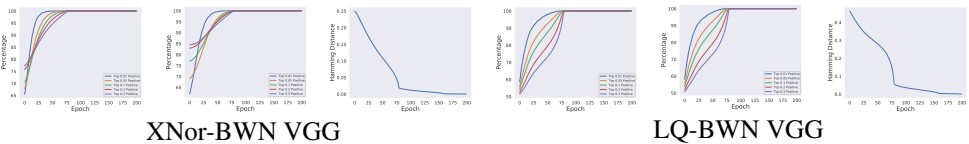

XNor-BWN VGG    LQ-BWN VGG

Figure 3: For each group, we display three figures whose X-axis is the training epoch and Y-axis is: **Left:** the overlapping percentage of those largest weights' signs between the final weights and the weights during training, **Middle:** the overlapping percentage of those largest weights' norm between the final weights and the weights during training, **Right:** the hamming distance divided by the number of parameters between binarized weights during training and the final trained binarized weights, which ranges between 1 (completely different from another) to 0 (the same).

network that has already finished training. One is to trace these large weights' signs, to find when these weights' signs become the same as the ones finishing training. Another is to trace these large weights' indices, to find when they become the largest weights among all weights.

The results of VGG-7 are shown in Figure.3. The results of ResNet-20 in Figure.9 and ResNet-18 in 10 are placed in Appendix.F. We find those large weights mostly have been decided in the early training stage. The larger magnitude these weights finally have, the earlier they decide and fix their sign. And this rule also applies to their magnitude, that the final trained weights with larger magnitude become having a larger magnitude in the very early stage. Both curves have a similar trend to the accuracy curve's trend.

## 2.5 PRIMARY BINARY SUB-NETWORKS IN BWNS

We find that there are weights with the large norm, fixing their signs in the early training stage. These weights are stable and vulnerable when inversing their signs. We name these weights as **Primary Binary Sub-Networks**. This idea is akin to the lottery ticket hypothesis (Frankle & Carbin, 2019), but the difference is our primary binary sub-networks' weights usually have fixed signs, and the rest of BWNs are not zero like the pruned networks. The primary binary sub-networks have the same norm for each weight after binarization, but different importance. **The lottery ticket is based on full-precision network pruning, and it pays more attention to getting sparse networks using the retraining method, while ours is to claim the meta idea that weights with larger norm are stable and sensitive on signs' changes. We will show how we utilize this idea in the rest paper.**

## 2.6 BINARY-KERNEL DISTRIBUTION

Besides the centered distribution of full precision weights in each layer, we find that there exists another distribution of binary-kernels in each layer. For a binary-kernel with $3 \times 3$ kernel size, there are $2^9$ possible kernels in total. For easier illustrations, we use 0 to 511 to index these kernels as shown in Figure.4. $3 \times 3$ kernels are more widely used in common CNN like VGG, ResNet, DenseNet (Huang et al., 2017), and MobileNet (Howard et al., 2017) (except the first Conv layer

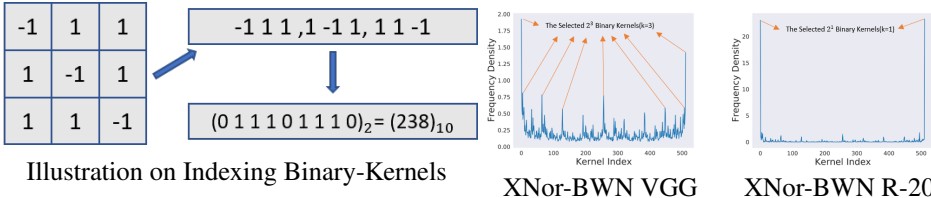

Illustration on Indexing Binary-Kernels       XNor-BWN VGG       XNor-BWN R-20

Figure 4: The visualization of binary weight kernels in one Conv layer after assigning indices. The X-axis indicates the index of a 3×3 binary weight kernel while Y-axis indicates the frequency that the binary kernel appears in one certain Conv layer. **Left Figure** is an example to illustrate how we index a $3 \times 3$ kernel into the range of 0 to 511. Two figures on the right are from the last Conv layer of two networks. **Right Figures** are the visualization of binary weight kernels in XNor-BWN VGG-7's last Conv layer and XNor-BWN ResNet-20's last Conv layer after assigning indices. The X-axis indicates the index of a $3 \times 3$ binary weight kernel while Y-axis indicates the frequency that the certain appears in Conv layer.

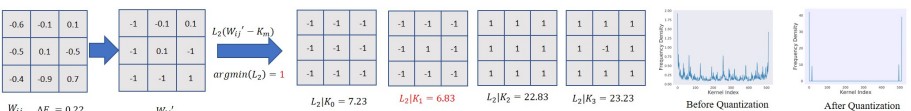

Figure 5: This is a pipeline to illustrate our compressing method on BWNs using 2-bit kernels. We first set those weights with larger norm into $\pm 1$ and keep those weights with the smaller norm, then calculate the $L2$ distance with 2-bit selected binary-kernels. After sorting the distances, we assign the one with the smallest distance to the original kernels. The right is two figures about the distribution of binary-kernels before and after quantization.

which is usually not binarized). From Figure.4, we can find that certain binary-kernels are in favor across different layers and networks.

## 3 QUANTIZED BINARY-KERNEL NETWORKS

In this section, we will introduce Quantized Binary-Kernel Networks (QBNs). In previous sections, we have several conclusions: 1. Scaling factors are not essential to BWNs, which guide us not to concentrate on designing scaling factors since good learning rates help in the most cases; 2. Weights with larger magnitude contribute to the primary binary sub-networks in BWNs, and these large weights are stable but sensitive on sign changing, determined and fixed in the early training stage; 3. Certain binary-kernels are centered on a limited number of the most frequent binary kernels. All these conclusions lead us to propose a new compression algorithm, which will further compress BWNs into a more structural and compact network, and we name this algorithm **Quantized Binary-Kernel Networks(QBNs)**. QBN basically is to the ultimate extent to maintain the primary binary sub-networks, changing those smaller weights' signs and quantize the less frequent kernels to those high frequent kernels to save space.

### 3.1 ALGORITHM

Before training a QBN, we first train an ordinary VGG-7 XNor-BWN on Cifar-10 and extract its last Conv layer's binary kernel distribution. This has been already done as shown in Figure.5. Then we sort these binary kernels according to their appearance frequency and select top $2^1, 2^2, ..., 2^8$ frequent binary kernels. These kernels are called selected binary-kernels $K_{0,1,...,2^8-1}$. In the rest of the paper, we use the selected binary kernels to indicate the kernels $K_{0,1,...,2^8-1}$ in our algorithm. In our following experiments these selected binary kernels are extracted from one single VGG-7 BWN's last Conv layer. After pre-processing these and obtaining $K_{0,1,...,2^8-1}$, we start to train a QBN using Algorithm.1, which is written with python-style pseudocodes. We use the function $where(A, B, C)$ from NumPy indicating that if the value satisfies A then equals to B otherwise equals to C.

---

**Algorithm 1:** QBN

---

Parameters: Quantized kernel bit number $p$, selected kernels $K_{0,1,...,2^p-1}$, hyper-parameter
  threshold $\Delta$, weight input channel number $I$, output channel number $O$, scaling factors
  $\alpha = 0.05$

Input: $W$

$E = \frac{\Sigma_{t=1}^{t=n}|W_t|}{n}$

**for** *w in W* **do**
   | **if** $\text{abs(w)} > \Delta E$ **then**
   |   | $w = \text{sign(w)}$
   | **end**
**end**

**for** *i in range (I)* **do**
   | **for** *j in range (O)* **do**
   |   | **for** *m in range ($2^p$)* **do**
   |   |   | $L_2(m) = ||W_{ij} - K_m||_2$
   |   | **end**
   |   | $m^* = \text{argmin}_m(L_2(m))$
   |   | $W_{ij} = \alpha K_{m^*}$
   | **end**
**end**

Return: W

---

Table 1: Experiments of VGG-7, ResNet-20, and ResNet-56 on Cifar10, and ResNet-18 on ImageNet. We put the results of baseline of full-precision networks and BWNs in Table.2. FP indicates the first full-precision Conv layer which is not quantized according to the common practice. VGG-7 has 6 Conv layers, and we use the quantized bit numbers for each layer to indicate how many selected quantized kernels are used. ResNet-20 and ResNet-56 have three groups, each group shares the same number of channels which are 16, 32, 64 in order. We assign the same quantized bit number for each group. ResNet-18 has four groups which have channels of 64, 128, 256, 512. CR indicates Compressed Ratio. Acc is the top-1 test accuracy on Cifar-10 and top-1 validation accuracy on ImageNet. The accuracy is reported as the mean of the best accuracy during training of 5 runs with different random seeds. More results are displayed in Appendix.H.

| Network | Quant Bit | CR | Acc | Network | Quant Bit | CR | Acc |
|---------|-----------|-----|------|---------|-----------|-----|------|
| VGG-7 | FP-6-5-4-3-2 | ×3.2 | 89.2% | VGG-7 | FP-3-3-3-3-3 | ×3.0 | 87.6% |
| R-20 | FP-9-9-1 | ×3.1 | 78.4% | R-20 | FP-9-9-2 | ×2.5 | 78.5% |
| R-56 | FP-5-5-5 | ×1.8 | 86.6% | R-56 | FP-9-6-2 | ×2.9 | 84.1% |
| R-18 | FP-4-4-4-4 | ×2.3 | 53.4% | R-18 | FP-9-7-4-1 | ×4.6 | 57.3% |

We set scaling factors fixed to 0.05 when using default learning rate mentioned in experimental settings of Section.2.2. We use $L_2$ norm to calculate the distance between the full-precision kernel $W_{ij}$ to the selected kernels $K_m$, where the full-precision kernel will be replaced by the selected kernel whose distance to the full-precision kernel is the shortest one during forward.

## 3.2 QBN EXPERIMENTS

We display our QBN experiments on in Table.1, where we use the same experiment settings mention in Section.2.2. Besides different networks and datasets are tested, we also use a different quantized bit on these networks to find how QBN can perform. When we use the quantized bit $p < 9$, we can use less than 9-bit number to represent the binary-kernel, this provides the compression ability of QBN. We use compressed ratio (CR) which is a number larger than 1 to show the ratio between the original BWNs and the compressed model's parameters only including binarized layers. In this paper, we do not use 8-bit quantized binary kernels, which have a high computational cost and small compressed ratio.

## 4 DISCUSSION ON QBN

In this section, we will discuss the experimental results of QBN and its potential usage, including model compression, kernel quantization strategies, the existence and transferability of the selected kernels, and other selection of binary-kernels.

### 4.1 MODEL COMPRESSION

With the discovery that BWNs contain primary binary sub-networks, we can reduce the number of parameters to represent a binary-kernel by changing the small magnitude weights' signs with bearable to the performance of BWNs. For VGG-7 on Cifar-10 and ResNet-18 on ImageNet, we can compress their parameters to an extremely small number by replacing the whole 512 types of 3×3 binary-kernel with fewer types of binary kernels from those $2^k$ selected binary-kernels, and the compressed ratio can be higher than $5\times$. For ResNet-20 and ResNet-56 which are thinner and have a small number of channels and parameters, they have a low endurance on compression, the compressed ratio can achieve to $1.5\times$ with a bearable accuracy drop (less than $3\%$ on Cifar-10). For a more aggressive compression with very low bit quantization binary-kernels, networks with fewer parameters like ResNet-20's training stability will drop due to their limited number of parameters. The experimental results are shown in Table.3 in Appendix.H.

### 4.2 CONNECTION BETWEEN PRIMARY BINARY SUB-NETWORKS

We use a hyper-parameter threshold $\Delta$ in Algorithm.1 to bridge QBN and Primary Binary Sub-Networks. When $\Delta = 0$, it means we first binarize all weights, then quantize these binary-kernels to those selected kernels. When $\Delta$ is large enough, it means we directly quantize the full-precision kernels. When $\Delta E$ is at a proper range of weight norm, those large weights will be first binarized to $\pm 1$. Considering the weight norm is usually a small value (from weight visualization in Figure.1 and Figure.8) compared to 1, these large weights receive a larger penalty by changing their signs during calculating the L2 distance between full-precision kernels and the selected binary-kernels. Thus, $\Delta$ is a hyper-parameter deciding how many portions will be considered as large weights, in the same term, Primary Binary Sub-Networks. According to our experiments of using different $\Delta$ in Figure.16 of Appendix.O, we find that $\Delta > 0$ is almost better than $\Delta = 0$. This sign() operation for all weights will eliminate the information of full-precision norm. Overall, these experimental results suggest our settings of $\Delta$ primary binary sub-networks first helping on quantizing binary-kernels, compared to binarizing weights first.

### 4.3 QUANTIZATION BIT STRATEGY

When using low quantization bit for binary-kernels, the performance drop will not be negligible, thus how to assign quantization bit to different layer is important. For VGG-7 and ResNet, they contain much more parameters in higher layers (layers near to the output), which have more channels, but their computational cost is similar in each layer. From the view of model compression, we find that higher layers have a higher endurance for the low-bit quantized kernels compared to lower layers (layers near to the input). Thus, we use low-bits in the last layer/group and use more-bits for the rest layers/groups to avoid bottlenecks.

### 4.4 EXISTANCE AND TRANSFERABILITY OF THE SELECTED KERNELS

To prove the existence of the selected kernels in other cases, and the transferability of our selected kernels, we did experiments on extracting top frequent kernels from different networks and layers and compare them with our selected kernels in Appendix.L. Then we conduct fine-tuning experiments for a pretrained BWN. This will be further studied in Appendix.M.

### 4.5 OTHER SELECTION OF BINARY-KERNELS

We discuss the other selection of binary-kernels in Appendix.N. For very low-bit quantization, we suggest using the most frequent binary-kernels rather than those less frequent ones. For the case like quantization bit $p > 4$, the choice of binary-kernels is not a essential problem.

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

## A  APPENDIX

In this appendix, we will display many additional experiments due to the limited pages in the Context. These experiments are all introduced in the Context as supplementary material to strength our main idea and make our contribution more convincing.

## B  EXPERIMENTS ON TRAINING BWNS

In this appendix section, we display our experiments on training different networks using XNor-BWN and LQ-BWN as shown in Table.2.

Table 2: This is a performance table of different quantization methods with different network architectures on different datasets with different training methods. "Baseline" means the network with full precision as the baseline. "w/o WD" means without weight decay, and "w/o SF" means without using scaling factors. $SF = 1$ means we fix scaling factors in all layers to 1. LR×10 means we magnify learning rate 10 times. Test Acc(Validation Acc on ImageNet) means top 1 accuracy on testing set(validation set).

| Network Methods | Test Acc | Network Methods | Test Acc |
|---|---|---|---|
| VGG-7 Baseline | 93.8% | | |
| VGG-7 XNor-BWN | 91.1% | VGG-7 LQ-BWN | 92.3% |
| VGG-7 XNor-BWN w/o WD | 90.9% | VGG-7 LQ-BWN w/o WD | 90.8% |
| VGG-7 XNor-BWN w/o SF | 91.0% | VGG-7 LQ-BWN w/o SF | 92.2% |
| VGG-7 XNor-BWN SF=0.05 | 90.3% | VGG-7 XNor-BWN SF=1 LR×20 | 89.7% |
| VGG-7 XNor-BWN SF=0.1 | 89.8% | VGG-7 XNor-BWN SF=1 LR×10 | 91.0% |
| VGG-7 XNor-BWN SF=1 | 85.1% | | |
| ResNet-20 Baseline | 91.3% | | |
| ResNet-20 XNor-BWN | 87.3% | ResNet-20 LQ-BWN | 88.6% |
| ResNet-20 XNor-BWN w/o WD | 86.0% | ResNet-20 LQ-BWN w/o WD | 87.4% |
| ResNet-20 XNor-BWN w/o SF | 87.0% | ResNet-20 LQ-BWN w/o SF | 88.5% |
| ResNet-18 Baseline | 69.6% | | |
| ResNet-18 XNor-BWN | 62.2% | ResNet-18 LQ-BWN | 61.1% |
| ResNet-18 XNor-BWN w/o SF | 62.5% | ResNet-18 LQ-BWN w/o SF | 62.4% |
| ResNet-18 XNor-BWN w/o WD | 61.0% | ResNet-18 LQ-BWN w/o WD | 58.3% |

## C  COMPARE LEARNABLE SF AND GAMMA IN BN

In original LQ-Net, they use an algorithm called 'Quantization Error Minimization' to calculate channel-wise quantizers, which are learnable scaling factors for each channel of Conv layers in binary weight case. Similarly, there is $\gamma$ in BatchNormalization layer, which also processes the pre-activation values in a channel-wise manner. In LQ-BWN, we normalize both $\gamma$ and scaling factors in such a channel-wise manner, and then plot the relation between these two values in Figure.6. As the figure shows, when the network goes deeper, two values behave highly correlated to each other, especially when the Conv layer is wide. Their high correlation between each other indicates the overlapping function of both.

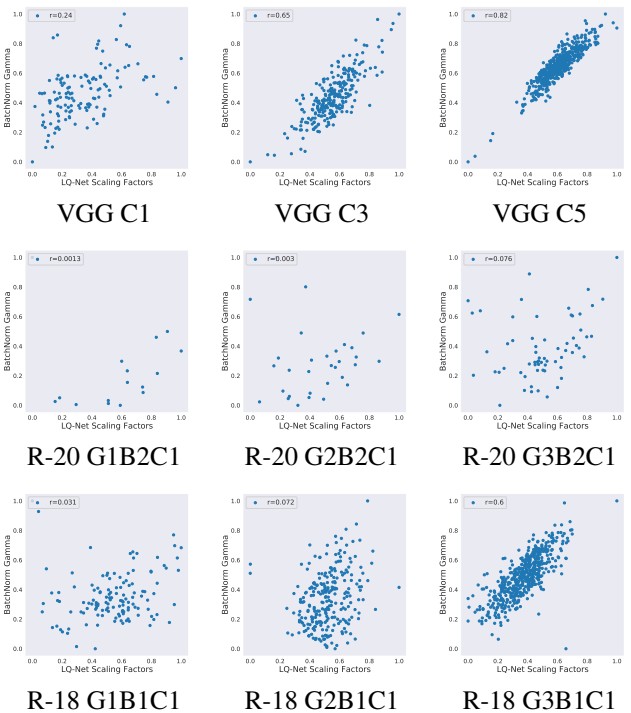

Figure 6: The relation between the scaling factor of the LQ-BWN and the gamma in BatchNorm layer after the corresponding Conv layer channel by channel after normalizing their values. The X-axis is the scaling factor value and Y-axis is the gamma in the corresponding BatchNorm layer. The point scattered in the figure is a combination of two values after normalization. **r** in the legend indicates the correlation coefficient. Texts below the figures indicates the certain layer, **VGG** is VGG-7, **R-20** is ResNet-20, **R-18** is ResNet-18, **C** is Conv, **G** is Group, and **B** is Block. These abbreviations will also be used in the rest of the paper.

## D    QUANTIZATION ERROR CURVE

In XNor-Net where XNor-BWN is first raised, the scaling factors are proposed to minimize the quantization error in a calculated deterministic way. In the BNN survey (Qin et al., 2020), the authors summarized several BWNs algorithms using "Minimizing the Quantization Error" which has a common form as shown in Equation3 to design their methods. Thus we plot the quantization error curve of different networks as shown in Figure.7. We can find that in most case the quantization error between full-precision weights and binary weights is not minimized. Therefore we are concerned that it might not be reasonable to use scaling factors to reduce the quantization error, even further, it might not be necessary to reduce the quantization error.

$$J(b, \alpha) = ||x - \alpha b||^2 \qquad \alpha^*, b^* = argmin_{\alpha, b} J(b, \alpha) \tag{3}$$

## E    WEIGHT HISTROGRAM ON OTHER NETWORKS AND LAYERS

We visualize the full-precision weight histrograms of ResNet-18's different layers in Figure.8.

## F    TRACING LARGE MAGNITUDE WEIGHTS

We trace the large magnitude weights' signs and their norm overlapping of ResNet-20 in Figure.9 and ResNet-18 in Figure.10. We also plot the hamming distance of ResNet-18 between the weights during training and weights after training in Figure.11. The experiment contents are the same as the ones in the Context.

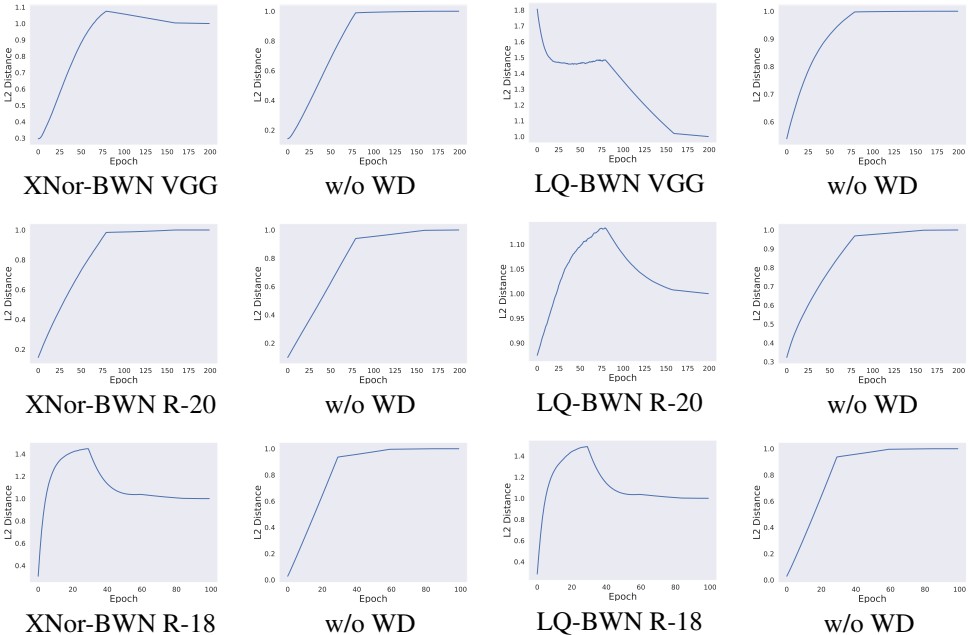

Figure 7: L2 Distance between full-precision weights and binarized weights during training. We use the L2 distance of the trained networks at last epoch as the unit norm, and other L2 distances are divided by this unit distance to better display the increase or decrease trend. textbfw/o WD means we do the same experiments on training the network without using weight decay. Two figures are displayed together and the left one uses WD while the right one does not. The X-axis is training epochs while Y-axis is the re-scaled sum of L2 norm of all binarized layers.

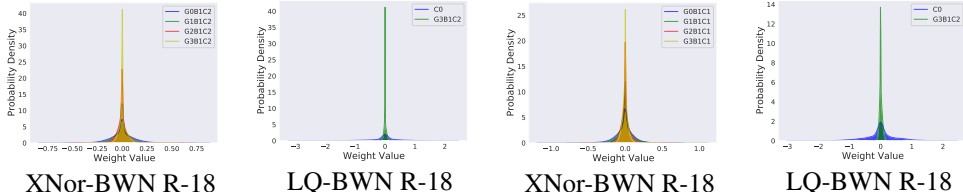

Figure 8: Full-precision weight histograms visualization in ResNet-18. We choose the weights from the second Conv in the first Block of each group, and the first Conv layer Conv0.

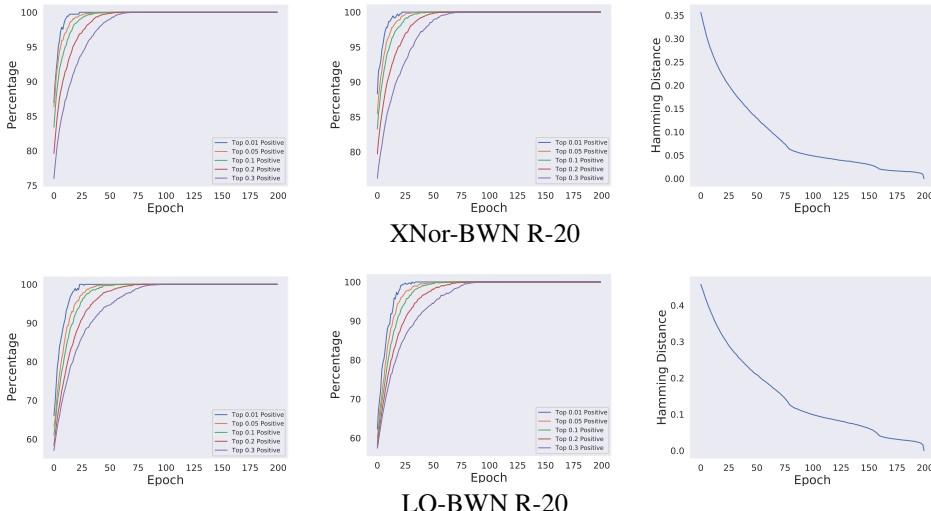

Figure 9: XNor-BWN and LQ-BWN ResNet-20 Weights' signs tracing and weights' norm overlapping with positive weights. And the hamming distance between the weights during training and the weights after training.

## G  FLIPPING AND PRE-FIXING LARGE MAGNITUDE WEIGHTS AND RETRAINING

We first flip 10% of weights of a trained BWN, and then fix its weights according to their magnitude and retraining it in Figure.12. The small learning rate in retraining is 0.0002, which is applied to the ordinary 160th to 200th epoch training. The large learning rate is using the original learning rate when training a BWN from scratch. We use three flipping strategy, flipping those largest magnitude weights, random weights, and those smallest magnitude weights.

## H  MODEL COMPRESSION ADDITIONAL EXPERIMENTS

We did more experiment with different quantization bit (together with different compressed ratio) on different networks and datasets as shown in Fig3.

## I  BINARY-KERNEL DISTRIBUTION IN EACH LAYER

We visualize the binary-kernel frequency of other layers and networks in Figure.13. We sum the frequency of top $2^p$ binary-kernels in each group of ResNet-18 with the log-scale x-axis.

## J  INSTABILITY TRAINING ON NETWORKS WITH VERY FEW PARAMETERS

When training QBNs on ResNet-20 with low bit quantizations, the test accuracy's variance is large, making training unstable. We write down a table on ResNet-20 with the training accuracy Standard Deviation among 5 times in Table.4.

## K  EXISTANCE AND TRANSFERABILITY OF THE SELECTED KERNELS

Selected binary-kernels do exist, and selected binary-kernels can transfer to other different layers, networks, or datasets. We extract different layers' top $2^1, 2^2, ..., 2^7$ frequent kernels percentage over all kernels and display in Table.13. It shows that usually, the imbalanced distribution among BWNs are very common, where only a small number of binary-kernels can take an advantage percentages

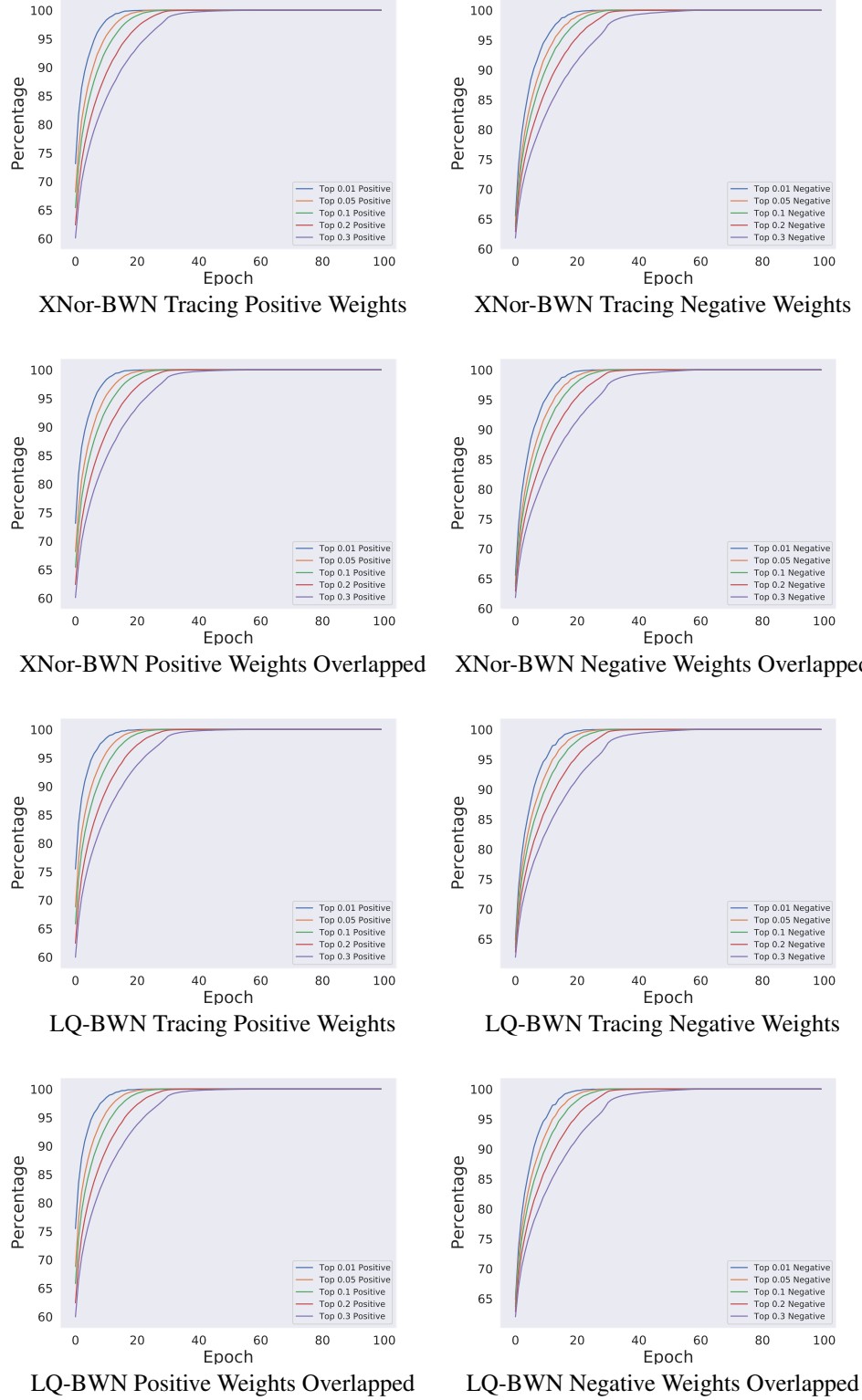

Figure 10: ResNet-18 Weights' signs tracing and weights' norm overlapping with both positive weights and negative weights.

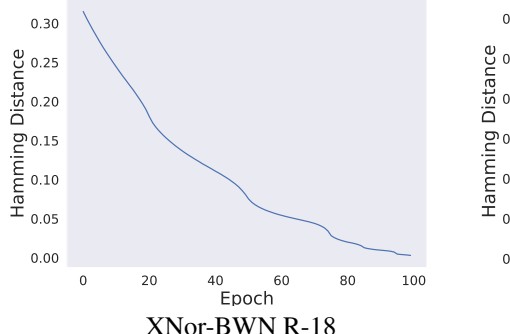
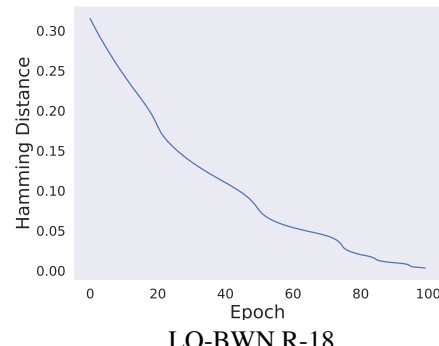

XNor-BWN R-18          LQ-BWN R-18

Figure 11: Hamming Distance between binary weights while training and binary weights after training. X-axis is the epoch number, and Y-axis is the hamming distance over the total binary weight numbers, zero means they become the same.

Table 3: Experiments of VGG-7, ResNet-20, and ResNet-56 on Cifar10, and ResNet-18 on ImageNet. FP indicates the first full-precision conv layer which is not quantized according to the common practice. VGG-7 has 6 conv layers, and we use the quantized bit numbers for each layer to indicate how many selected quantized kernels are used. ResNet-20 and ResNet-56 has three groups, each groups share the same number of channels which are 16, 32, 64 in order. We assign the same quantized bit number for each group. ResNet-18 has four groups which has channels of 64, 128, 256, 512. CR indicates Compressed Ratio. Acc is the top 1 test accuracy on Cifar-10, and top 1 validation accuracy on ImageNet. The accuray is reported as the mean of the best accuracy during training among 5 runs with different random seeds.

| Network | Quant Bit | CR | Acc | Network | Quant Bit | CR | Acc |
|---------|-----------|-----|------|---------|-----------|-----|------|
| VGG-7 | FP-9-9-9-9-9 | ×1.0 | 92.2% | VGG-7 | FP-2-2-2-2-2 | ×4.5 | 84.8% |
| VGG-7 | FP-6-5-4-3-2 | ×3.2 | 89.2% | VGG-7 | FP-3-3-3-3-3 | ×3.0 | 87.6% |
| VGG-7 | FP-4-4-4-4-1 | ×3.7 | 89.0% | VGG-7 | FP-3-2-2-1-1 | ×7.2 | 88.9% |
| VGG-7 | FP-5-4-3-2-1 | ×4.9 | 89.0% | VGG-7 | FP-5-5-5-5-5 | ×1.8 | 91.3% |
| R-20 | FP-9-9-9 | ×1.0 | 87.6% | R-20 | FP-5-3-1 | ×5.8 | 63.4% |
| R-20 | FP-9-9-1 | ×3.1 | 78.4% | R-20 | FP-9-9-2 | ×2.5 | 78.5% |
| R-20 | FP-9-9-3 | ×2.0 | 80.6% | R-20 | FP-4-4-4 | ×2.2 | 63.8% |
| R-20 | FP-5-5-5 | ×1.8 | 64.3% | R-20 | FP-6-6-6 | ×1.5 | 81.0% |
| R-20 | FP-9-6-3 | ×2.3 | 75.8% | R-20 | FP-6-4-2 | ×3.5 | 62.8% |
| R-20 | FP-9-4-1 | ×4.7 | 64.6% | | | | |
| R-56 | FP-9-9-9 | ×1.0 | 89.2% | R-56 | FP-9-6-2 | ×2.9 | 84.1% |
| R-18 | FP-9-9-9-9 | ×1.0 | 64.5% | R-18 | FP-3-3-3-3 | ×3.0 | 49.3% |
| R-18 | FP-4-4-4-4 | ×2.3 | 53.4% | R-18 | FP-9-7-4-1 | ×4.6 | 57.3% |
| R-18 | FP-5-5-5-5 | ×1.8 | 56.6% | R-18 | FP-9-6-4-2 | ×3.4 | 57.5% |
| R-18 | FP-6-6-6-6 | ×1.5 | 59.0% | R-18 | FP-9-9-6-2 | ×2.8 | 60.2% |
| R-18 | FP-7-7-7-7 | ×1.3 | 62.6% | | | | |

Table 4: Experiments of QBN on ResNet-20. We use (mean ± std) to report the results.

| Network | Quantization Bit | Acc |
|---------|------------------|------|
| ResNet-20 | FP-5-3-1 | (63.4± 5.6)% |
| ResNet-20 | FP-4-4-1 | (65.9± 2.7)% |
| ResNet-20 | FP-6-6-3 | (70.5± 3.1)% |
| ResNet-20 | FP-9-6-3 | (75.8± 2.4)% |
| ResNet-20 | FP-9-9-1 | (78.4± 1.4)% |
| ResNet-20 | FP-9-9-2 | (78.5± 2.7)% |
| ResNet-20 | FP-9-9-3 | (80.6± 0.9)% |
| ResNet-20 | FP-9-9-9 | (88.0± 0.8)% |

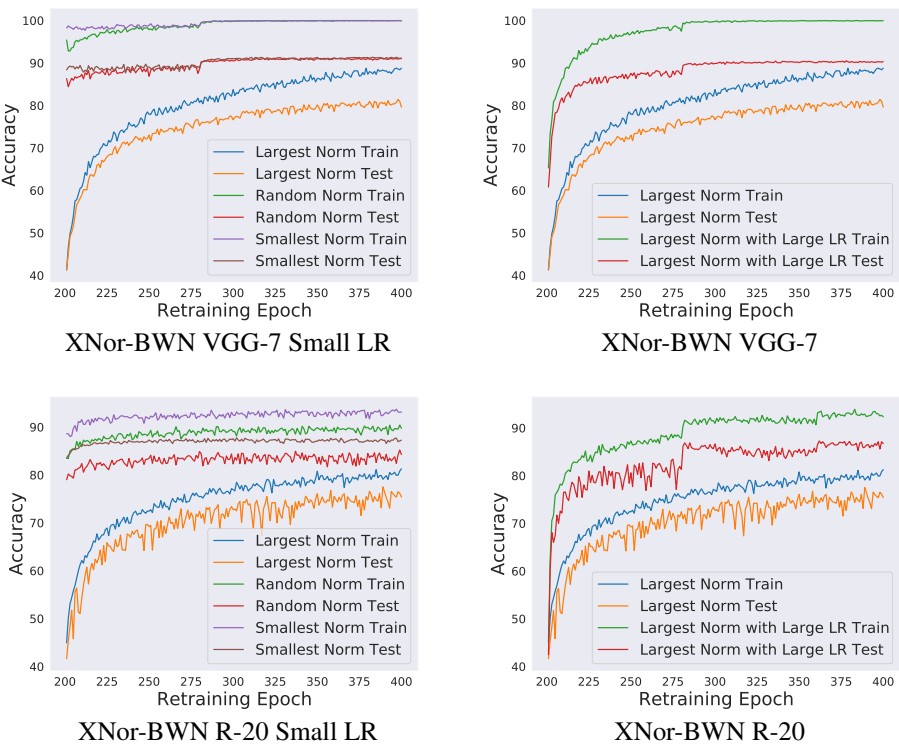

Figure 12: Flip 10% of a trained BWN and fixing these weights, retrain with different learning rates, on VGG-7 and ResNet-20. The left two figures compare the flipping strategies, and the right two figures compare the learning rate magnitude, which use small learning rate and large learning rate.

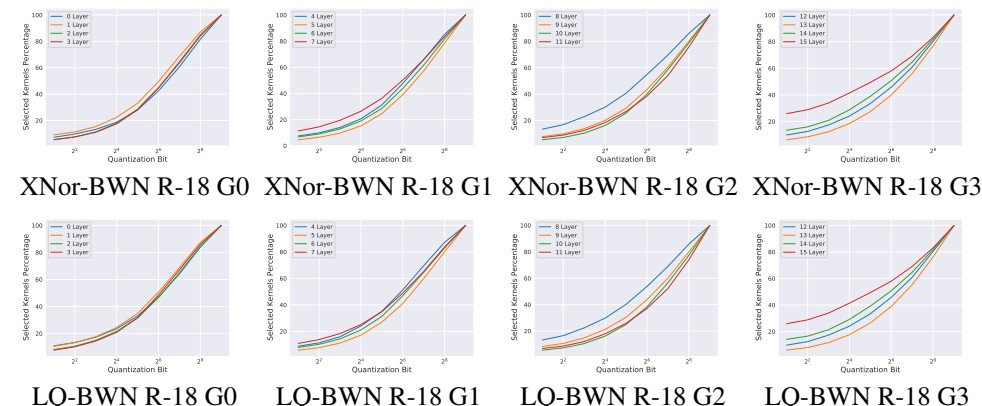

Figure 13: The sum of frequency of top $2^p$ binary-kernels of XNor-BWN and LQ-BWN ResNet-18. X-axis is log2 scale to visualize quantization bits. G0 to G3 are the groups of ResNet-18, and each group contains two blocks, two layers for each block.

of all binary kernels. This proves that the QBNs algorithm can adapt to a wide range of networks and datasets.

To prove the transferability of the selected binary-kernels, Table.3 shows that the experiments are all based on the same batch of the selected kernels, which are extracted from the last Conv layer of a VGG-7 XNor-BWN without any additional tricks. The results on ResNet-20, ResNet-56 on Cifar-10, and ResNet-18 on ImageNet with such residual structures and different layers show how

those selected kernels perform well. We display how the selected binary-kernels we use throughout our experiments from the last Conv of VGG-7 appear in other networks' different layer, plotting a Figure.14 to compare their frequency.

We also apply QBNs to fine-tune pre-trained BWNs, when using relatively more quantization bit, the network can usually gain comparable performance without training from scratch. In our QBN algorithm, the computational cost increases when using more quantization bits, thus we can directly fine-tune a pre-train BWN. The experiments in Appendix.M show that we can gain a uniform $7 - bit$ QBN in one epoch, and a uniform $6 - bit$ QBN in two epochs. This provides a higher efficient way to train those high quantization bit QBNs.

## L  THE DEGREE OF AGGREGATION OF BINARY-KERNELS

We do a statistic on how much percentage of the target networks' top frequent binary-kernels can be covered by our selected binary-kernels from the last Conv of XNor-BWN VGG-7. We use the first figure on the top left of Figure.14 to demonstrate that, when using $1 - bit$ quantization on binary-kernels of XNor-BWN ResNet-18's first 4 layers(the first group of ResNet-18), our selected kernels can cover 100% of XNor-BWN ResNet-18's top $2^1$ frequent kernels. The percentage has already been processed with the number of each kernel in the layer. 100% is the limit that if we directly choose the most frequent kernels from this layer/group as our selected kernels. For the lowest percentage appearing when using 5-bit, there are still more than 80% kernels in the first group's top $2^5$ frequent kernels can be represented directly by our selected kernels.

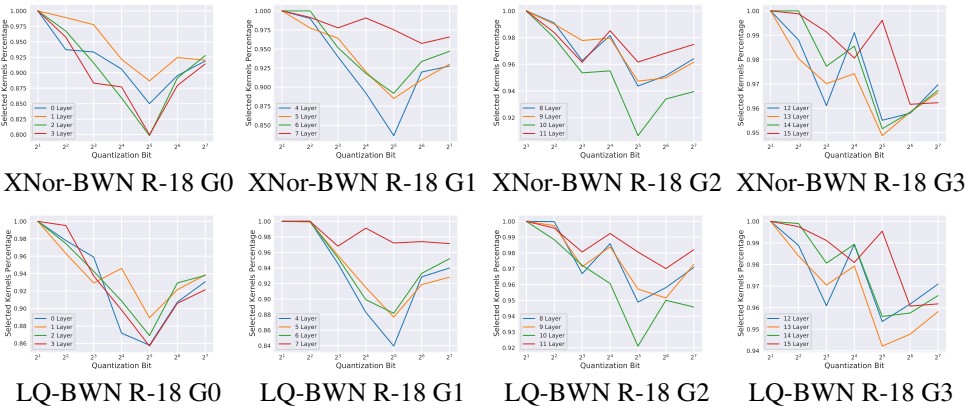

Figure 14: The ratio between the number of kernels in one layer that are in our selected binary-kernels, and that are in the top $2^p$ frequent binary-kernels in that layer. Thus 100% is the upper bound, which means our selected binary-kernels can cover the same number of kernels that this layer's top frequent kernels can cover. The X-axis is log2 scale to visualize quantization bits. G0 to G3 are the groups of ResNet-18, and each group contains two blocks, two layers for each block.

## M  FINE-TUNE WITH HIGHER QUANTIZATION BIT(ON 6 BIT, 7 BIT)

We write a Table.5 using relatively higher quantization bit to fine-tune ResNet-18 on ImageNet with one epoch. The pretrained model is the normal ResNet-18 BWN at 90th epoch. We list the number of epochs that the fine-tune requires. For those lower bit, the fine-tuned performance cannot achieve to the one training from scratch.

## N  OTHER SELECTION OF BINARY-KERNELS

Here we list three reasons to illustrate why we use such a way to collect the binary-kernels. **1.** If we regard QBNs as a clustering method given cluster numbers, these kernels with high appearance frequency are most likely the cluster centers. **2.** We visualize these top frequent kernels as shown

Table 5: ResNet-18 on ImageNet funetuned base on a pretrained BWN.

| Network | Quantization Bit | Acc |
|---------|------------------|-----|
| ResNet-18 | FP-9-9-9-9 | 62.93%(90 Epochs) |
| ResNet-18 | FP-6-6-6-6 | 58.79%(90 Epochs + Retrain 2 Epochs) |
| ResNet-18 | FP-6-6-6-6 | 58.91%(90 Epochs + Retrain 10 Epochs) |
| ResNet-18 | FP-7-7-7-7 | 62.28%(90 Epochs + Retrain 1 Epoch) |
| ResNet-18 | FP-7-7-7-7 | 62.56%(90 Epochs + Retrain 10 Epochs) |

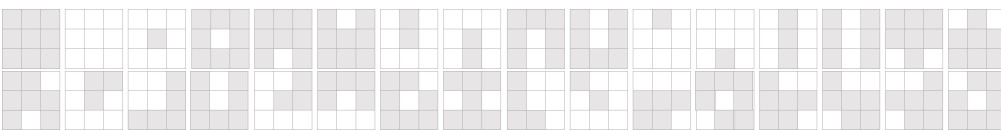

Figure 15: **Top:** The visualization of Top 16 frequent binary-kernels. **Down:** The visualization of the least frequent 16 binary-kernels among top $2^7$ frequent binary-kernels, more specifically, the 113rd to 128th frequent kernels.

in Fig.15, and they appear similar to conventional spatial filters, including "Moving Average Filter", "Point Detector", "Line Detector" and so on. **3.** We use the less frequent binary-kernels as the selected kernels to test if our selection based on frequency is a good choice. We find that using those less frequent kernels, an accuracy drop can be observed in different experiments. Those less frequent kernels are used by inverting the order of top 128 frequent kernels, eg. for $2^2$ kernels, they are 125-th, 126-th, 127-th, and 128-th frequent kernels. Given the experiment results in Table.6, we find that when using very low quantization bit, specifically less than or equal to 3, will significantly hurt the network when using those least frequent kernels. When the quantization bit increases, the difference in performances reduce. Thus we suggest to use those most frequent kernels if training a very low-bit QBN, or finetuning a pre-trained BWN. In other cases like training from scratch with quantization bit more than 3, the frequency of the selected kernels is not a strict deterministic factor.

Table 6: We train several networks using the normal top frequent binary-kernels, and those least frequent binary-kernels named "Reverse" in the table since they use the order of reversed top 128 frequent kernels. $10.0\%$ in the table indicates the training cannot converge from the beginning.

| Network and Quantization Bit | Acc(Normal) | Acc(Reverse) |
|------------------------------|-------------|--------------|
| VGG-7 FP-3-3-3-3-3 | 87.0% | 10.0% |
| VGG-7 FP-4-4-4-4-4 | 89.7% | 90.7% |
| VGG-7 FP-5-5-5-5-5 | 90.5% | 92.1% |
| ResNet-20 FP-9-6-3 | 75.9% | 74.6% |
| ResNet-20 FP-9-9-2 | 78.9% | 75.0% |
| ResNet-20 FP-9-9-3 | 80.6% | 78.6% |
| ResNet-56 FP-9-6-2 | 84.1% | 81.4% |
| ResNet-56 FP-5-5-5 | 86.6% | 87.0% |

## O INFLUENCE OF HYPER-PARAMETER $\Delta$

To test the influence of the Hyper-Parameter $\Delta$ in our algorithm, we use VGG-7 with all 5-bit quantization kernels and ResNet-56 with FP-9-6-2. We do not use ResNet-20 due to its instability training which we have discussed. The result is shown in Figure.16.

## P THE SELECTED KERNELS FROM OTHER CONV LAYERS

In previous experiments, our selected binary kernels $K_m$ use the statistic information extracted from one single VGG-7's last Conv layer and regard them as constant kernels. We further use different strategies to demonstrate that the selected binary kernels have a strong generalization ability to apply on other networks. In the following experiments, we use the selected binary kernels from the last

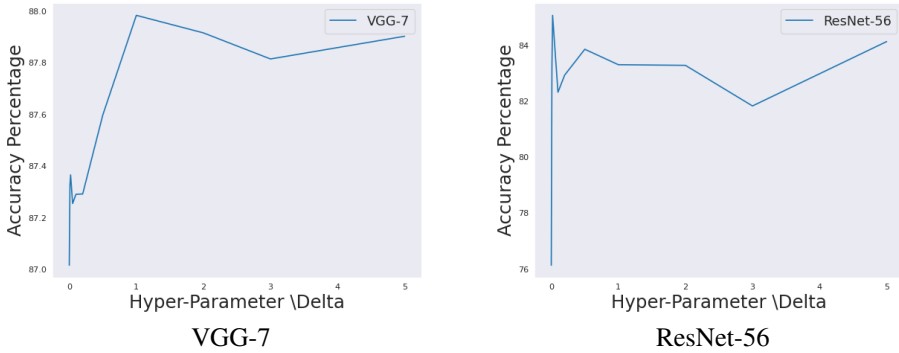

VGG-7                    ResNet-56

Figure 16: The test accuracy of 5 experiments for each hyperparamter $\Delta$ of different values.

layer of ResNet-20, VGG-7's 3rd and 5th Conv layers, and from the last layer of ResNet-18. We use these kernels from different strategy to other different networks. The result is shown in Table.7.

Table 7: Using different source layers of the selected kernels. The Source Layer is where we extract the most frequent binary kernels, and target QBN is the network we apply QBN to.

| Source Layer | Target QBN | Quant Bit | Acc |
|---|---|---|---|
| VGG-7 3rd Layer | VGG-7 | FP-5-5-5-5-5 | 91.0% |
| VGG-7 3rd Layer | VGG-7 | FP-6-5-4-3-2 | 89.0% |
| VGG-7 3rd Layer | ResNet-20 | FP-9-9-3 | 79.4% |
| VGG-7 3rd Layer | ResNet-20 | FP-6-6-6 | 81.0% |
| VGG-7 5rd Layer | VGG-7 | FP-5-5-5-5-5 | 91.2% |
| VGG-7 5rd Layer | VGG-7 | FP-6-5-4-3-2 | 89.2% |
| VGG-7 5rd Layer | ResNet-20 | FP-9-9-3 | 80.6% |
| VGG-7 5rd Layer | ResNet-20 | FP-6-6-6 | 81.4% |
| ResNet-18 13th Layer | VGG-7 | FP-5-5-5-5-5 | 90.9% |
| ResNet-18 13th Layer | VGG-7 | FP-6-5-4-3-2 | 88.9% |
| ResNet-18 13th Layer | ResNet-20 | FP-9-9-3 | 79.0% |
| ResNet-18 13th Layer | ResNet-20 | FP-6-6-6 | 81.4% |
| ResNet-18 17th Layer | VGG-7 | FP-5-5-5-5-5 | 91.0% |
| ResNet-18 17th Layer | VGG-7 | FP-6-5-4-3-2 | 89.2% |
| ResNet-18 17th Layer | ResNet-20 | FP-9-9-3 | 79.8% |
| ResNet-18 17th Layer | ResNet-20 | FP-6-6-6 | 81.9% |

