# OpenReview forum: "Weights Having Stable Signs Are Important: Finding Primary Subnetworks and Kernels to Compress Binary Weight Networks"
_ICLR.cc/2021/Conference — Reject_

### Official Review · AnonReviewer3 · 2020-10-14
**This idea is novel and experiment results are supervising. I tend to accept this paper.**

**Rating:** 6
**Confidence:** 4

**Review:**

Overview:
The Authors show that scaling factors with hand-crafted or learnable methods are not so important when training Binary Weight Networks (BWNs), while the change of weight signs is crucial. They make two observations: The weight signs of the primary binary sub-networks are determined and fixed at the early training stage. Binary kernels in the convolutional layers of final models tend to be centered on a limited number of fixed structural patterns. Based on these observations, they propose a new method called binary kernel quantization to further compress BWNs.

Strength bullets:
1. They propose a binary kernel quantization to quantize the binary kernel, which effectively reduces the number of all possible kernels, and can further compress BWNs to 2-5 times.
2. The Authors observe that the weight with the large norm, fixing their signs in the early training stage. And they compared this phenomenon with the lottery ticket hypothesis and propose the primary binary sub-networks. I think it's an interesting idea.
3. The paper is well written and well-motivated. It contains extensive and systematic related work-study. I like it.

Weakness bullets:
1. The binary kernel quantization method is only applied to 3x3 kernels in this paper. When the size of the convolution kernel is 5x5 or 7x7，how does it work? I hope this method can extend to another cases and it also can work well.
2. As shown in table1, why authors report the best accuracy among different seed? I think authors should report the averaged results with different random seeds instead of the best result.

-----Post Rebuttal-----

I have read all feedback and especially thanks to the authors' efforts on the extra experiments. I think the author has addressed my first concern. For the second one, I would prefer to see the errorbar.

I tend to keep my scores unchanged. I think the findings are interesting [share similar thoughts as R4's], while the experiments part need to be improved. Although I like the ideas and observations, I don't feel especially strongly in favor of it and cannot champion it.

---

> ### Author Response · Authors · 2020-11-25
> **To AnonReviewer3 “This idea is novel and experiment results are supervising. I tend to accept this paper.”**
>
> Thank you so much for the thoughtful review and the recognition of our work. Please see our below responses to your questions.
> 1.	**Questions**: “The binary kernel quantization method is only applied to 3x3 kernels in this paper. When the size of the convolution kernel is 5x5 or 7x7,how does it work? I hope this method can extend to another cases and it also can work well.”
>
> **Answers**: **(1)** On the one side, in the neural network quantization field, it is a common benchmarking protocol, retaining the first convolutional layer of a CNN model to still have full-precision values. On the other side, to the prevailing CNNs such as VGG, ResNets and MobileNets, large convolutional kernels like 5x5 or 7x7 are merely adopted in their first convolutional layer while the other convolutional layers all use 3x3 (2^9 binary patterns in total) or 1x1 (2 binary patterns) convolutional kernels. For a fair experimental study and comparison, in the previous version of our submission, we followed this protocol when applying the proposed method (i.e., Quantized Binary-kernel Networks) to compress binary weight networks. This is why our method is only applied to 3x3 kernels in the experiments and analysis (it was stated in Section 2.6); **(2)** Following your suggestion, we further applied our method to the first layer (with 7x7 convolutional kernel) of ResNet-20 as well as the other layers. Comparatively, for 7x7 convolutional kernel the number of possible binary patterns is 2^49 which is significantly larger than 2^9 for 3x3 convolutional kernel, and we observed most binary patterns only appear once or do not appear, leaving the flexibility to determine primary patterns. As a result, good compression results are also obtained and displayed as the followings:
>
> Network               --- Quant Bit --- Acc
>
> ResNet-20(64C5) ---  25-9-9-9   ---  83.1%
>
> ResNet-20(64C5) ---  7-7-7-7     ---  78.2%
>
> ResNet-20(64C5) ---  7-6-6-6     ---  77.5%
>
> ResNet-20(64C7) ---  49-9-9-9   ---  83.8%
>
> ResNet-20(64C7) ---  7-7-7-7     ---  77.6%
>
> ResNet-20(64C7) ---  7-6-6-6     ---  75.9%
>
>
> 2.	**Questions**: “As shown in table1, why authors report the best accuracy among different seed? I think authors should report the averaged results with different random seeds instead of the best result.”
>
> **Answers**: It is a typo that we report the mean of 5 random runs’ best accuracy during training. We have corrected this description to ‘the mean of the best accuracy during training among 5 runs with different random seeds’.

---

### Official Review · AnonReviewer2 · 2020-10-15
**Official Blind Review #2**

**Rating:** 3
**Confidence:** 5

**Review:**

**Summary:**
This paper proposes some interesting observations for training BWNs. 1: The scaling factors can be removed with batch normalization used. 2: The signs of the weights with large norms are determined and fixed at the early training stage. 3:  The binary weight networks can be further compressed. Moreover, the authors provide some empirical visualizations and results to demonstrate its analysis. However, the paper seems to be incomplete and needs to be further improved.

**Pros:**
The observation that the signs of weights with large norms are determined and fixed at the early training stage is interesting. The idea that weights with large norms are stable and sensitive on sign changes can be utilized for improving the training of BWNs, maybe BNNs as well.

**Cons:**
1: Extensive descriptions are very confusing. I can only list some of them.

1): The contributions of the paper include the observation that binary kernels in the learned convolutional layers tend to be centered on a limited number of fixed structural patterns. However, it is still not clear to me what the “fixed structural patterns” are.

2): In Sec. 3, the paper claims to “quantize the less frequent kernels to those high frequent kernels to save space” and “we sort these binary kernels according to their appearance frequency…” . However, the paper fails to explain the motivation for exploring the frequency of kernels. Some theoretical explanations are needed.

3):  In Figure 4’s caption, what does “the certain appears in one certain Conv layer” mean?

4):  In Sec. 4.1, the authors propose to apply QBN on model compression.  However, it is not clear to me what the points the paper intends to express. Specifically, the authors claim that “we can reduce the number of parameters to represent a binary-kernel by changing the small magnitude weights’ signs”. Then the question comes. How can the number of parameters be reduced via changing signs? Also, the sentence “we can compress their parameters to an extremely small number by replacing the whole 3×3 binary-kernel with fewer”  is incomplete.

2: Extensive symbols are undefined, which makes me hard to understand the paper properly, especially in Algorithm 1, the main algorithm of the paper. For example, definition of $n$ in calculating $E$;  definition of  “where()”;  definition of $K_m$, are not explained.

3:  Extensive technical details are missing, which makes the algorithm difficult to understand.
1):   The authors should at least explain how to generate the power-of-two number of binary kernels in details in Sec. 3.1.
2):   During training, the paper generates $k$-bit (i.e., k is the bitwidth) number of binary kernels. However, during testing, the learnt binary weights should be fixed ($k=1$) and why the “Quant Bit” in Table 1 can be larger than 1? This makes me very confusing.

4:  This paper only binarizes weights while I am wondering whether simultaneously binarizing both weights and activations can have the similar observations.

5:  There is no comparisons with current state-of-the-arts.  Also, in Sec 4.1, the paper claims that “the compressed ratio can achieve to 1.5$\times$ with a bearable accuracy drop (less than 3% on Cifar-10)” does not stand. The 3% loss on Cifar-10 is significant.

6:  There is no conclusions and future works discussions.

7:  It is widely known that the scaling factors can be absorbed into BN layers. In terms of this, it should not be a contribution of this paper.

8: The paper introduces extensive heuristic hyper-parameters. Specifically, it does not give any strategy to automatically determine the optimal quantized bitwidth and threshold $\Delta$ for each layer.

9:  There are lots of grammar and typo issues. For example, the words in equations should use straight format (use \rm in latex).

---

> ### Author Response · Authors · 2020-11-25
> **To AnonReviewer2 “Official Blind Review #2.” (Part 1)**
>
> Thank you so much for the thoughtful review and the recognition of our work. Please see our below responses to your questions.
>
> 1: Extensive descriptions are very confusing. I can only list some of them.
>
> **Questions**:  1): The contributions of the paper include the observation that binary kernels in the learned convolutional layers tend to be centered on a limited number of fixed structural patterns. However, it is still not clear to me what the “fixed structural patterns” are.
>
> **Answers**:  The fixed structural patterns are those most frequent patterns, more specifically, those 3x3 binary kernels that appear more frequently than other types of binary kernels in BWN. One example is that the binary kernels with all +1 or all -1 appear much more frequently than other kernels as we have shown in Figure 4. To make our description clearer, we have changed the term ‘fixed structural patterns’ to ‘the most frequent binary kernels’.
>
> **Questions**:  2): In Sec. 3, the paper claims to “quantize the less frequent kernels to those high frequent kernels to save space” and “we sort these binary kernels according to their appearance frequency…” . However, the paper fails to explain the motivation for exploring the frequency of kernels. Some theoretical explanations are needed.
>
> **Answers**:  As shown in Figure 4 about the frequency of total 512 types of binary kernels, they are highly unevenly distributed. To compress them which means using fewer types of binary kernels, we need to find cluster centers of these binary kernels. Thus, we use a straightforward method, directly regarding those binary kernels appearing most frequently as the cluster centers.
>
>
> **Questions**:  3): In Figure 4’s caption, what does “the certain appears in one certain Conv layer” mean?
>
> **Answers**:  We correct this caption to “one binary kernel appears in one certain Conv layer.”
>
>
> **Questions**:  4): In Sec. 4.1, the authors propose to apply QBN on model compression. However, it is not clear to me what the points the paper intends to express. Specifically, the authors claim that “we can reduce the number of parameters to represent a binary-kernel by changing the small magnitude weights’ signs”. Then the question comes. How can the number of parameters be reduced via changing signs?
>
> **Answers**:  (1) We illustrate how we did model compression beyond BWN in Figure 5. By choosing four binary kernels, we assign all 512 types of binary kernels into one of these four binary kernels. Thus one original 3x3 binary kernel which requires a 9-bit binary code to represent only needs a 2-bit binary code instead since we can use such 2-bit binary code to indicate any binary kernels after applying QBN in BWN. The right figures in Figure 5 shows the binary kernel frequency distribution’s changes before and after using QBN. (2) Our findings in Section 2 that weights with smaller norm bring less influence on BWN compared to those weights with larger norm guarantee that we can cluster those binary kernels by changing those small weights’ signs.
>
> **Questions**: Also, the sentence “we can compress their parameters to an extremely small number by replacing the whole 3×3 binary-kernel with fewer” is incomplete.
>
> **Answers**:  We have rewritten this sentence to “we can compress their parameters to an extremely small number by replacing the whole 512 types of 3×3 binary-kernel with fewer types of binary kernels from those 2^k selected binary-kernels” to make it more clear.
>
>
> 2: Extensive symbols are undefined, which makes me hard to understand the paper properly, especially in Algorithm 1, the main algorithm of the paper.
>
> **Questions**: For example, definition of in calculating ; definition of “where()”; definition of , are not explained.
>
> **Answers**:  In the algorithm we use the python-style pseudocodes to explain our method. We have replaced the definition of “where()” by “if-then”.
>
>
> 3: Extensive technical details are missing, which makes the algorithm difficult to understand.
>
> 1): **Questions**: The authors should at least explain how to generate the power-of-two number of binary kernels in details in Sec. 3.1.
>
> **Answers**:  We have answered this question in 1. 4), and in Sec. 3.1. we have explained that these power-of-two number of kernels are selected from one single VGG-7’ last Conv layer’s top 2^1, 2^2, …, 2^8 frequent binary kernels.
>
> 2): **Questions**: During training, the paper generates k-bit (i.e., k is the bitwidth) number of binary kernels. However, during testing, the learnt binary weights should be fixed (k=1) and why the “Quant Bit” in Table 1 can be larger than 1? This makes me very confusing.
>
> **Answers**:  In Figure 4, we display how to represent a 3x3 binary kernel which is represented by a 9-bit binary code. And in Figure 5, we show how a binary kernel can be assigned to fewer binary kernel which only needs a binary code with k-bit (k<9) to represent.  We use k-bit to represent how many quant bits we use in each layer/block.

---

> ### Author Response · Authors · 2020-11-25
> **To AnonReviewer2 “Official Blind Review #2.” (Part 2)**
>
> 4: **Questions**: This paper only binarizes weights while I am wondering whether simultaneously binarizing both weights and activations can have the similar observations.
>
> **Answers**: (1) Currently we only focus on the case which the weights are binarized; (2) Following your suggestion, we applied our QBN on BNN (1-bit weight and 1-bit activation). On VGG-7, using Quant Bit FP-5-5-5-5-5 drops 4.2% compared to using 32-bit activation, using Quant Bit FP-4-4-4-4-4 drops 5.6% compared to using 32-bit activation.
>
> 5: **Questions**: There is no comparisons with current state-of-the-arts. Also, in Sec 4.1, the paper claims that “the compressed ratio can achieve to 1.5 with a bearable accuracy drop (less than 3% on Cifar-10)” does not stand. The 3% loss on Cifar-10 is significant.
>
> **Answers**: (1) We focus on deeper understanding and analyzing BWN and propose the QBN algorithm to prove the correctness of our findings, providing a possible application based on these findings. Achieving or comparing with the current state-of-the-arts is not our main purpose in this paper. Besides our BWN baselines are using the original algorithm from the paper ‘XNor’ in 2016, and so far there is no similar work like ours; (2) The performance drop becomes relatively large on ResNet-20 and ResNet-56 which have smaller parameter numbers compared to VGG-7 on Cifar-10. This is caused by ResNet-20 and ResNet-56 architectures on Cifar-10. We further discuss applying QBN on networks with very small number of parameters like ResNet-20 brings the performance drop in Table 4 in Appendix. On VGG-7, we can reach the compression ratio 1.8 with 0.9% accuracy drop. Moreover, we did not stress that our main contribution is getting a state-of-the-arts compression strategy, but rather the observations, analyzing based on experiments, and the prototype algorithm. Improving the accuracy will be our future works.
>
>
> 6: **Questions**: There is no conclusions and future works discussions.
>
> **Answers**:  Our paper is written in an experiment-driven manner. The method and experiments are about the discussion. Due to the page limit, we did not give a separate section to summarize the conclusion and future work. We would add them in the future if more space is available.
>
> 7: **Questions**: It is widely known that the scaling factors can be absorbed into BN layers. In terms of this, it should not be a contribution of this paper.
>
> **Answers**:  (1) We agree absorbing scaling factors into BN layers in inference or deploying section is a common practice and we have already mentioned this in our. (2) However, our proof of Equation 1 and 2 is to prove that scaling factors are already absorbed by BN and can be removed when using BN *during training*. Besides, we do not need to retrain one epoch to absorb the scaling factors or other operations. (3) In previous work, XNor claimed that using a calculated optimal scaling factor in BWN plays an important role for training BWN. We also use experiments to demonstrate that simply using 1 as scaling factors during training and inference can also achieve the comparable performance as shown in Table 2.
>
> 8: **Questions**: The paper introduces extensive heuristic hyper-parameters. Specifically, it does not give any strategy to automatically determine the optimal quantized bitwidth and threshold for each layer.
>
> **Answers**:  (1) As we have mentioned in our abstract, our QBN algorithm is to testify our hypothesis, that changing those signs of weights which have smaller norm brings less performance drop, and binary kernels are centered on several patterns and can be clustered to be compressed further. (2) We agree that in QBN where we use heuristic hyper-parameters such as threshold ∆ and binary kernels bit-width for each layer, however, we insist that QBN is a prototype application based on our findings and understandings of BWN. Either achieving states-of-the-art or making it automatically work is not our primary purpose (3) We believe the QBN algorithm has a large potential to be improved on its performance and made into AutoML to better deployed in reality in the future.
>
> 9: **Questions**: There are lots of grammar and typo issues. For example, the words in equations should use straight format (use \rm in latex).
>
> **Answers**:  We did proof reading and corrected the improper grammars and typos in our paper.

---

### Official Review · AnonReviewer1 · 2020-10-28
**Extensive empirical results about BWN**

**Rating:** 5
**Confidence:** 2

**Review:**

Summary:
The authors show empirically that weight signs are more important than weight magnitudes. They also analyze the optimization process and the structure of optimized binary networks to note that
i) there are clusters of weights whose weight remain almost unchanged during the optimization
ii) optimized convolutional networks have simple sub-structures
They exploit the latter to propose a new quantization method that increases the compression of binary networks.

strengths:
The idea of studying how binary weights update during the optimization is interesting and may help understand the optimization process of binary but also real-valued networks. The experiments include numerical results for a wide range of data sets and networks.

weaknesses:
The meaning of the given simple proof is not clear. What are gamma and epsilon in equations 1 and 2? The paper would have much more impact if, after showing that BN can absorb the redundant factors, all real-valued parameters were dropped. In some sense, the proof seems to show that the obtained result about the little influence of scaling is somehow expected.

Regrading the sign flipping, observing that "flipping weights with large full precision magnitude will cause a significant performance drop compared to those weights close to zero" seems also something that one can expect.

Finally, the presence of clusters is not really explained and shown in a very accessible way. Are there other results except from Figure 4 (whose caption sounds a bit hermetic: "The X-axis indicates the index of a 3 x 3 binary weight kernel while Y-axis indicates the frequency that the certain appears in one certain Conv layer")

questions:
- what happens if all scaling factors are kept fixed?
- is alpha usually shared by all weights in the network or is layer-specific?
- the weights distributions in figure 1 are obtained by adding a regularization term?
- what happens if also the first full-precision Conv layer is quantized?

---

> ### Author Response · Authors · 2020-11-25
> **To AnonReviewer1 “Extensive empirical results about BWN.”**
>
> Thank you so much for the thoughtful review and the recognition of our work. Please see our below responses to your questions.
>
> 1.	**Questions**: The meaning of the given simple proof is not clear. What are gamma and epsilon in equations 1 and 2? The paper would have much more impact if, after showing that BN can absorb the redundant factors, all real-valued parameters were dropped. In some sense, the proof seems to show that the obtained result about the little influence of scaling is somehow expected.
> **Answers**: (1) In equations 1 and 2, we use the same notations of all parameters of original BN. Gamma and Beta are the affine parameters of BN while epsilon is a parameter for avoiding dividing zero when doing standardization (PyTorch uses 5e-4 manually). \bar{x} and \sigma are the estimated moving average of mean and variance which are calculated during training and keep fixed after finishing training. We do not need to retrain one epoch to absorb the scaling factors or other operation. (2) In additional, we fixed scaling factors \alpha = 1 rather than use the sum(|w|)/n which is used in XNor-BWN in Table 2. We achieve a similar result using scaling factors \alpha= 1 and XNor-BWN’s scaling factors. This may address your concern that all real-valued parameters were dropped.
>
>
> 2.	**Questions**: Regrading the sign flipping, observing that "flipping weights with large full precision magnitude will cause a significant performance drop compared to those weights close to zero" seems also something that one can expect.
> **Answers**:  To best of our knowledge, this is the first work studying the influence of full precision weight magnitude on their corresponding binary weight’s sign providing with extensive experiments to prove the correctness of this observation.
>
> 3.	**Questions**: Finally, the presence of clusters is not really explained and shown in a very accessible way. Are there other results except from Figure 4 (whose caption sounds a bit hermetic: "The X-axis indicates the index of a 3 x 3 binary weight kernel while Y-axis indicates the frequency that the certain appears in one certain Conv layer")
> **Answers**:  (1) In Figure 4, in order to display the frequency of all 512 types of binary kernels, we first reshape 3x3 kernels to encode each of them into a 9-bit binary code, then convert them into a integer between 0 and 511. This helps us to visualize the frequency distribution. (2) We correct the caption to “the binary kernel appears in one certain Conv layer.”
>
>
> For questions:
> 1.	**Questions**: what happens if all scaling factors are kept fixed?
> **Answers**:  In our experiments, we use scaling factors \alpha = 0.05 for all binarized layers and networks. We also give the results that when scaling factors \alpha = 1, making learning rate 10 times larger brings the similar performance in the Table 2.
> 2.	**Questions**: is alpha usually shared by all weights in the network or is layer-specific?
> **Answers**:  In our experiments, the scaling factor alpha is the same for all networks and all layers. But for other works, cases are different. XNor-BWN use layer-specific alpha, and LQ-BWN use a learnable channel-specific alpha.
> 3.	**Questions**: the weights distributions in figure 1 are obtained by adding a regularization term?
> **Answers**:  The weights distributions are extracted from BWNs which use weight decay as the only regularization term.
> 4.	**Questions**: what happens if also the first full-precision Conv layer is quantized?
> **Answers**:  It is a common practice in BWN that we leave the first layer and the last layer as full precision weights. Here we also apply our QBN that binarizes the first Conv layer. The result is shown in Table 8 in the Appendix, please see “Summary of Changes in the New Version”.

---

### Official Review · AnonReviewer4 · 2020-11-04
**Interesting findings but needs more clarity and stronger results**

**Rating:** 5
**Confidence:** 4

**Review:**

This paper provides an empirical study of binary weight networks (BWNs), where they find that 1 the commonly adopted scaling factor is not critical 2 there exists a subnetwork that stabiles early in training 3 the 3x3 filters in VGG and ResNets demonstrate a sparse distribution. They combine all the observations and propose a novel quantization algorithm that achieves more aggressive compression than standard BWNs.

pros:
+ I appreciate the careful examination of design and training details of standard BWNs. The identification of a persistent subnetwork and the analysis on the sparse distribution of kernels are particularly interesting.

+ The proposed quantization algorithm is interesting, which has a potential of squeezing more redundancy out of standard BWNs

cons:
- If I understand correctly, in the proposed algorithm the kernel distribution is only drawn from the last conv layer of the full precision network, which is then shared across all layers when retraining the BWN. This seems a strong assumption and needs to be justified. What's the reason to believe that the selected frequent kernels are shared across different layers?

-In Algorithm 1, W = where(abs(W ) > ∆E, sign(W ), W ) is not motivated and explained well. What's the reasoning of using the threshold when computing the distance to the frequency binary kernels?

-The experimental results seem to be really hard to interpret for me, and this is perhaps the weakest point of the this paper. In particular, Table 1 needs to have proper baselines. This includes the full precision, standard BWN accuracies, as well as controls which allow one to draw comparisons between the proposed algorithm and basic binarization by equating certain quantities.

I suggest the authors work on the suggested improvements which will make this a much stronger contribution.

*****post rebuttal updates*****
I want to thank the authors for responding to my questions. The additional explanations are indeed helpful for clarifying my first two questions (selection of the binary kernel and the use of ∆E). However, I still have concerns about Table 1 (and Table 2). For example, I have a really hard time interpreting the significance of achieving a 3.2x CR with a loss of 3% (92.3 - 89.2 from VGG-7) in acc with the proposed method (although the paper argues that it's a "bearable" loss). Considering that this is the main experiment supporting the efficacy of the proposed quantization algorithm, I think the paper needs more controlled experiments to demonstrate the practical usefulness of the proposed algorithm. As a result I'm keeping my original score and hope the authors can work on the improvements for the next version.

---

> ### Author Response · Authors · 2020-11-25
> **To AnonReviewer4 “Interesting findings but needs more clarity and stronger results.”**
>
> Thank you so much for the thoughtful review and the recognition of our work. Please see our below responses to your questions.
> 1.	**Questions**: If I understand correctly, in the proposed algorithm the kernel distribution is only drawn from the last conv layer of the full precision network, which is then shared across all layers when retraining the BWN. This seems a strong assumption and needs to be justified. What's the reason to believe that the selected frequent kernels are shared across different layers?’
>
> **Answers**:  Though it contains a strong assumption that these selected binary kernels can generalize across different networks, different layers, and different datasets, we give two reasons for the justification:
>
> (1)	We did an experiment to discover to what extend do the binary kernel distributions between VGG-7 BWN’s last layer’s and other different networks’ different layers’ in our Appendix L. In Figure 14, we show that the selected binary kernels from one single-trail VGG-7 BWN’s last layer are highly corelated to those most frequent binary kernels in ResNet-18’s different layers. For example, for the same k-bit kernels, the top 2^k frequent binary kernels contain a% of all binary kernels in one layer, and the selected 2^k binary kernels from VGG-7 BWN’s last layer contain b% (b<=a because a% is already the highest percentage that 2^k binary kernels can achieve). So, we use b/a <= 1 to represent such correlation.  b/a = 1 indicates that the top 2^k frequent binary kernels are identical to our selected 2^k binary kernels. In the figures, all are higher than 80%, and for the deeper layers (the last two blocks in ResNet-18) they are higher than 90%. This indicates that those most frequent kernels are similar across different networks and different layers and different datasets.
> (2)	Besides the correlation experiments, we also choose the selected kernels from the 2nd layer of VGG-7, from 4th layer of VGG-7, from the last conv layer of ResNet-18, from the 9th conv layer of ResNet-18(the last conv of the second block of ResNet-18), and merging the statistic information of these two layers of ResNet-18. Then we apply these new selected binary kernels to VGG-7 on Cifar-10 as shown in Table.7.
>
> 2.	**Questions**: In Algorithm 1, W = where(abs(W ) > ∆E, sign(W ), W ) is not motivated and explained well. What's the reasoning of using the threshold when computing the distance to the frequency binary kernels?
> **Answers**:  In Section 4.2, with the experiments we discuss about ‘Connection between Primary Binary Sub-Networks’. For this ∆E, we give the explanation that this step is to make those large weights more important when calculating the L2 by assigning them with a larger number +-1 instead of their original norm(the norm of weights are much smaller compared to 1). In the paper we did further experiments on such threshold in Appendix O with Figure 16, which indicates that ∆=0 is worse in both VGG-7 and ResNet-56 with the worse performance compared to other cases that ∆>0. For ∆>0 the optimal value for ∆ is different case by case.
>
> 3.	**Questions**: The experimental results seem to be really hard to interpret for me, and this is perhaps the weakest point of the this paper. In particular, Table 1 needs to have proper baselines. This includes the full precision, standard BWN accuracies, as well as controls which allow one to draw comparisons between the proposed algorithm and basic binarization by equating certain quantities.
> **Answers**:  Due to the page limit, in the paper we put the full-precision network and BWN baselines in Table 2 in Appendix. Follow your suggestion, we refer the results of baseline of full-precision network and BWN in Table 1 caption to make it easier to follow.

---

### Author Response · Authors · 2020-11-17
**Rebuttal and paper revision are in preparation**

We sincerely appreciate all reviewers for their thorough and constructive comments. To address the concerns and requests from reviewers, we are carefully improving the presentation, experiments, and discussions. Our detailed rebuttal responses, as well as a paper revision will be submitted in the following week.

We thank all reviewers again for their time, feedbacks and patience.

---

### Author Response · Authors · 2020-11-25
**General Response**

Thanks for four reviewers’ hardworking. Following the advice and the raised questions, we update our paper in the following four aspects:

**1**. We update our description on QBN algorithm to make it easier to follow.

**2**. We rewrite the caption in Figure 4 which led a misunderstanding to the reviewers. And we also update the right figure in Figure 4 to make it more accessible.

**3**. We add new experiments in Appendix and responses as suggested by reviewers.

**4**. We check the grammars, typos, and other details.


*Updated Paper*:

Abstract

* As suggested by Reviewer2, we use ‘the most frequent binary kernels’ instead of ‘fixed structural patterns’ which may cause misunderstanding.

Section2
* As suggested by Reviewer1, in Subsection3 Scaling Factors, we add the notation of Gamma and Epsilon in Equation1 and 2 to make it more accessible.

* As suggested by Reviewer1 and 2, we rewrite the caption of Figure 4 and add the descriptions on right two figures for more clear illustration.

Section3
* As suggested by Reviewer1, in Subsection1 Algorithm, we re-write the description on “the selected binary-kernels” and emphasize that we use “one single VGG-7 BWN's last Conv layer” to obtain our selected binary-kernels in our following experiments.

* As suggested by Reviewer2, we use if-else function to replace where() function to make our algorithm more clear.

* As suggested by Reviewer4, in our Table 1 caption, we add ‘We put the results of baseline of full-precision networks and BWNs in Table.2.’ to refer the baselines.

* As suggested by Reviewer3, in our Table 1 caption, we use ‘the mean of the best accuracy during training among 5 runs with different random seeds’ to report our results.

Appendix

* Table 7: As suggested by Reviewer4, we add the new table that using different sources of selected binary kernels to different networks to demonstrate the selected frequent kernels can generalize across different networks and datasets.

---

### Decision · Program_Chairs · 2021-01-07
**Final Decision**

**Decision:**

Reject

**Comment:**

## Description
The paper discovers interesting phenomena in training neural networks with binary weights:
- Connection between latent weight magnitude and how important its binarized version for the network performance
-training dynamics, indicating that large latent weights are identified and stabilize early on
- Observation that amongst learned binary kernel, several specific patterns prevail, up to the bits who's reversal has very little effect. This is so regardless of the architecture, the layer considered or the dataset.
The paper further demonstrates how these observations may be used to compress binary neural networks below 1 bit per weight.

## Review Process and Decision
The reviewers welcomed the experimental investigation of new phenomena, but commented the overall technical quality of the work as somewhat substandard. The redundancy of consecutive affine transforms is known and not connected to binary weights investigation. The investigation itself lacks a more in-depth analysis. The proposed compression results appeared not convincing to reviewers since a significant drop of accuracy occurs. The AC shares these concerns and supports rejection.

## General Comments
From my perspective, the study undertaken is methodologically „wrong“. An ad-hoc training method is investigated, which is not even clearly defined in the paper (there are many „STE“ variants) and for which it is not known what it is doing, what are the real-valued weights for and whether they are needed at all (as empirically argued by Helwegen et al. (2019)). As such, the investigation makes impression of poking a black box (the training method in this case). At the same time, there are more clear learning formulations, applicable in the setting of the paper (binary weights), in particular considering the stochastic relaxation:
* Shayer et al. (2017): Learning Discrete Weights Using the Local Reparameterization Trick
* Roth et al. (2019): Training Discrete-Valued Neural Networks with Sign Activations Using Weight Distributions
* Peters et al. (2018): Probabilistic binary neural networks

These methods are approximate, but at least the optimization is well posed and it is known what do the real-valued weights represent (e.g. logits of binary weight probabilities).
From this perspective, it can be seen that latent weights close to 0 correspond to Bernoulli weights that are almost fully random (and thus only contribute noise) and are fragile to gradient steps. Therefore the model can only perform well if it learns to be robust to their state or their state becomes more deterministic (corresponding to large latent weight). So one would actually expect to see in these models phenomena similar to the observed in the paper and not bee too much surprised or astonished by them. Furthermore, there are recent works explaining STE and its latent weights as optimizing the stochastic relaxation:
* Meng et al. (2020): Training Binary Neural Networks using the Bayesian Learning Rule
* Yanush et al. (2020): Reintroducing Straight-Through Estimators as Principled Methods for Stochastic Binary Networks.

The authors are encouraged to make the observed phenomena more explainable by connecting to the mentioned works.

## Further Details

*  „We show that in the context of using batch normalization after convolutional layers, adapting scaling factors with either hand-crafted or learnable methods brings marginal or no accuracy gain to final model.“

From theoretical perspective, this is obvious and known to me. Practically, there could be in principle some difference due to the learning dynamics, and verifying that there is none is a useful but a weak contribution. The section devoted to this issue can be given in the appendix but is not justified in the main paper.

* „change of weight signs is crucial in the training of BWNs“

The sign determines the binary weights, so this is by definition.

* „ Firstly, the training of BWNs demonstrates the process of seeking primary binary sub-networks whose weight signs are determined and fixed at the early training stage, which is akin to recent findings on the lottery ticket hypothesis for efficient learning of sparse neural networks“

In the lottery ticket hypothesis paper it is shown explicitly that the identified sparse subnetwork changes during the learning the most rather than retains its initialization state or the state in the beginning of the training. It is therefore could be of a different nature.